# Twentieth-Century Paleoproteomics: Lessons from Venta Micena Fossils

**DOI:** 10.3390/biology11081184

**Published:** 2022-08-06

**Authors:** Jesús M. Torres, Concepción Borja, Luis Gibert, Francesc Ribot, Enrique G. Olivares

**Affiliations:** 1Departamento de Bioquímica y Biología Molecular III e Inmunología, Universidad de Granada, 18016 Granada, Spain; torrespi@ugr.es (J.M.T.); conchaborja56@gmail.com (C.B.); 2Departament de Geoquímica, Petrologia i Prospecció Geològica, Universitat de Barcelona, 08028 Barcelona, Spain; lgibert@ub.edu; 3Museo de Prehistoria y Paleontología Josep Gibert, 18858 Orce, Spain; fribottrafi@hotmail.es; 4Instituto de Biopatología y Medicina Regenerativa, Centro de Investigación Biomédica, Universidad de Granada, 18100 Armilla, Spain; 5Unidad de Gestión Clínica Laboratorios, Hospital Clínico Universitario San Cecilio, 18016 Granada, Spain

**Keywords:** fossil proteins, ELISA, paleoproteomics, RIA, Venta Micena site, VM-0, VM-1960

## Abstract

**Simple Summary:**

Two independent research groups led by Olivares (Spain) and Lowenstein (USA) investigated the immunological reactions of proteins extracted from the controversial Orce skull (VM-0), a 1.3-million-year-old fossil found at the Venta Micena site in Orce, Granada (Spain) and initially believed to come from an unidentified hominin. Work by both groups with polyclonal and monoclonal antibodies showed that proteins from this fossil reacted most strongly to antibodies against modern human proteins. Other hominin and mammal fossils from Venta Micena were also studied.

**Abstract:**

Proteomics methods can identify amino acid sequences in fossil proteins, thus making it possible to determine the ascription or proximity of a fossil to other species. Before mass spectrometry was used to study fossil proteins, earlier studies used antibodies to recognize their sequences. Lowenstein and colleagues, at the University of San Francisco, pioneered the identification of fossil proteins with immunological methods. His group, together with Olivares’s group at the University of Granada, studied the immunological reactions of proteins from the controversial Orce skull fragment (VM-0), a 1.3-million-year-old fossil found at the Venta Micena site in Orce (Granada province, southern Spain) and initially assigned to a hominin. However, discrepancies regarding the morphological features of the internal face of the fossil raised doubts about this ascription. In this article, we review the immunological analysis of the proteins extracted from VM-0 and other Venta Micena fossils assigned to hominins and to other mammals, and explain how these methods helped to determine the species specificity of these fossils and resolve paleontological controversies.

## 1. Introduction

All living organisms carry their own evolutionary history in their cells, and this history can be read in analyses of nucleic acid sequences or protein amino acid sequences. Thus, phylogenetic trees constructed from DNA or proteins have helped to clarify evolutionary relationships among species. Although DNA and proteins are also determinants of morphology, the genetic information that morphology provides is indirect and difficult to interpret, since numerous genes and complex interrelations are involved in configuring the structures of a living organism. In addition, convergence or parallel evolution phenomena can lead to similarities between unrelated species in one or more morphological characteristics. Nevertheless, in classical paleontology, species identification and classification are based exclusively on morphological features of the fossil record. As Wilson and Cann remarked, “The fossil record, on the other hand, is infamously spotty because a handful of surviving bones may not represent the majority of organisms that left no trace of themselves. Fossils cannot, in principle, be interpreted objectively: the physical characteristics by which they are classified necessarily reflect the models the paleontologists wish to test” [1]. Alongside morphological features, the analysis of biomolecules that survive in fossils can be of great help in identifying and classifying these remains, especially when they are fragmented and their morphological classification is controversial. Molecular paleontology methods have been developed to deal with this challenge.

## 2. Short Survival of Fossil DNA, Longer Survival of Fossil Proteins

Although recent decades have seen spectacular developments in molecular paleontology, this branch of science is not as recent as has been suggested. In the 1950s, Abelson first demonstrated the presence of amino acids and peptides in fossils [2]. In their initial work the detection of amino acids present only in certain proteins (e.g., hydroxyproline in collagen) made it possible to infer which type of protein these amino acids came from, but no additional genetic information could be obtained regarding the species to which the rest of the amino acids belonged. In 1963, Wykoff published electron microscopy images of collagen fibrils in dinosaur bones more than 200 million years old—another example of the early stages of molecular paleontology [3]. The molecule that has most often been investigated in the tissues of extinct species, ancient bones, or fossil remains is DNA, as the direct carrier of genetic information. The first ancient DNA sequence was obtained by Wilson’s group, who studied a museum specimen of tissue from a quagga—a species from the horse family that became extinct in the late 19th century. To study the DNA remnants in the sample it was necessary to amplify them with a technique first developed in 1984: molecular cloning [4].

The main drawback that limits the scope of molecular paleontology is that any biomolecules that may survive in fossil remains must necessarily be altered and present at very low concentrations. When an organism dies, most of its biomolecules, as well as the organism itself, quickly disappear. However, under special circumstances in which rapid dehydration or rapid burial in an anaerobic environment occurs, hard (bone, shell, etc.) and even soft tissues (skin, muscle, etc.) can survive, and may thus contain biomolecules—albeit not in an intact form [5]. Proteins found in fossils are thus usually denatured and fractionated into peptides. In addition, after death, a process of racemization takes place: amino acids with the *L*-form spatial configuration are converted to the isomeric *D*-form [6]. DNA, an even more fragile molecule than proteins, is usually fractionated into sequences of only a few hundred base pairs containing abundant lesions such as baseless sites, oxidized pyrimidines, and chain cross-linkings [7]. Accordingly, Lindahl noted that it would be unlikely that any useful DNA could ever be extracted from very ancient fossils [8]. In fact, although the entire Neanderthal genome has been sequenced [9], most studies that focused on DNA found that it is unlikely to survive for more than 100,000 years. Nonetheless, notable exceptions to date are the sequencing of this biomolecule in a 400,000 year old *Homo heidelbergensis* fossil [10], and the genomic data obtained from a 560–780-thousand-year-old equid specimen [11] and from two mammoth specimens more than 1 million years old [12]. In addition, Woodward et al. published nine DNA sequences of the gene encoding cytochrome b, which were extracted from an 80-million-year-old dinosaur bone [13]. A drawback of these results was that the sequences did not show a significant degree of similarity to equivalent sequences from birds and reptiles, i.e., dinosaurs’ closest extant relatives. However, later analyses of the sequences obtained by Woodward and colleagues revealed a greater similarity to human DNA than to that of other animals. This similarity, therefore, ruled out the possibility of dinosaur DNA and showed that the results were probably due to the inadvertent contamination of the sample during processing [14].

Although the ideal outcome is to read genetic information directly from the DNA nucleotide sequence, proteins also provide useful information, albeit indirectly, about amino acid sequences. In contrast to DNA, which appears to survive for only a thousand years, some proteins, under certain conditions, can persist in fossils for millions of years. Proteins bind to the mineral phase (hydroxyapatite) of bone, and this binding provides considerable protection from degradation by exogenous agents. Moreover, the amount of hydroxyapatite crystals increases after death, and this may favor protein encapsulation [15,16]. Compared to DNA, however, proteins present a technical obstacle in that they are not amplifiable, so their concentration cannot be increased—as can be attempted for DNA with polymerase chain reaction techniques. Although initial studies conducted between the 1950s and 1970s identified amino acids and peptides in fossils up to millions of years old, they did not provide information on the species specificity of these biomolecules, that is, on their ascription to or kinship with other species [2,3].

## 3. Detection of Fossil Proteins with Immunological Methods: Applications in Paleontological Controversies

Proteins undergo profound changes over time; nonetheless, these molecules, although fragmented or altered, can in some cases retain intact amino acid sequences. The protein fragments may be detectable with antibodies, which identify sequences comprising between 4 and 12 amino acids (epitopes) [17]. Mass spectrometry (MS) is also able to detect amino acid sequences in peptides [18]; however, this technology had not yet been implemented for fossil proteins in the twentieth century. In this period, most studies that aimed to identify fossil proteins were carried out with antibodies. Jerold M. Lowenstein was the first to identify genetic information contained in fossil proteins by applying radioimmunoassay (RIA) [19], an immunological technique able to specifically detect proteins in quantities as low as 10^−13^ M. Lowenstein and colleagues found human collagen, the most abundant protein in bone, in fossil samples of 20,000-year-old *Homo sapiens*, 50,000-year-old *Homo neanderthalensis*, 0.5-million-year-old *Homo erectus*, and 1.9-million-year-old *Australopithecus robustus* [20,21]. Collagen was also detected with dot-blotting in a 10-million-year-old fossil bone [22]. Osteocalcin, another abundant protein in bone, was identified by Ulrich et al. with antibodies in 13-million-year-old fossil bovine bones and 30-million-year-old rodent teeth. These researchers observed that osteocalcin in bovine fossil material still retained its functional ability to bind calcium [23]. Osteocalcin was also detected in a sample of 75-million-year-old dinosaur bone [24].

Particularly interesting is the detection of proteins in *Ramapithecus* fossils. In the 1960s, some paleoanthropologists considered this species, which lived 8 to 20 million years ago, to be a hominid, and it was thus suggested that the human lineage had diverged from that of apes about 20 million years ago. Molecular data, however, contradicted this hypothesis. Sarich and Cronin used immunological techniques to study modern chimpanzee, gorilla and human albumin, and concluded that these three species diverged from a common ancestor only 5 million years ago [25]. If this hypothesis is correct, it rules out hominin ancestry for *Ramapithecus*. Lowenstein produced antibodies by injecting an extract prepared from this fossil into a rabbit, and found that these antibodies reacted more strongly with gorilla, orangutan and gibbon sera than with chimpanzee or human sera. According to these results, *Ramapithecus* was genetically as closely related to Asian monkeys as to African monkeys, and more distantly related to humans [26]. Currently, paleontologists do not include *Ramapithecus* in the human lineage and consider it more closely related to orangutans.

At the turn of the twentieth century, a skull of modern human appearance was discovered in Sussex, England, which appeared in association with a jaw displaying ape-like morphological characteristics. Because the morphology of these bones was consistent with then-current theories of human evolution, the so-called Piltdown Man (*Eoanthropus dawsoni*) was accepted in 1912 by English anthropological authorities as the missing link between apes and humans, and was considered the first English human. It was not until 1953 when it became evident, based on an analysis of fluoride content, that the purported fossil was a fraud: a 500-year-old human skull to which the artificially aged jaw of a monkey had been added and the teeth modified to make them look human [27]. It remained to be determined whether the jaw was from a chimpanzee or orangutan. Lowenstein et al. studied a sample from the jaw and observed that antibodies to orangutan collagen reacted more strongly with an extract from this sample than did antibodies to human or chimpanzee collagen [28].

## 4. The Case of Orce Man

### 4.1. The Orce fossils

The Orce fossils assigned to hominins by Josep Gibert and colleagues include a skull fragment (the so-called Orce skull, VM-0), a humeral diaphysis (VM-1960), and a distal fragment of humerus (VM-3691) [29]. These remains were found at the Venta Micena site near the town of Orce in the province of Granada, southern Spain. The age of this site has been estimated magnetostratigraphically as 1.3 million years; if accurate, this would make the fossils the oldest evidence of a hominin presence in Europe. It has been hypothesized that Orce Man colonized Europe from the south by crossing the Strait of Gibraltar [30]. There is currently general agreement that early humans occupied the Orce area between 1.3 and 1.2 million years ago, based on a limited number of stone tools and on evidence of anthropic actions on bones detected at Venta Micena quarry 3 [31]. Additional evidence has come from nearby sites: two human deciduous molars, probably belonging to the same individual, were discovered at the Barranco Leon site [32,33,34], and stone artefacts were found at the Fuentenueva-3a and Barranco León-5 sites [35,36]. However, as often occurs in paleoanthropology, the Orce fossils became the subject of intense controversy [37]. In contrast to the position of Gibert and colleagues [38], some Spanish paleontologists claimed that the Orce skull belonged to an equid or a ruminant, and that the humeri were too incomplete to be identified with certainty [39,40]. Nevertheless, reputable paleoanthropologists such as Phillip V. Tobias, after close examination of the fossils assigned to hominins, supported Gibert’s conclusions [41]. Given the uncertainties surrounding the morphological features, immunological studies of the fossil molecules were undertaken.

### 4.2. Methods to Investigate Proteins in Fossils

The Venta Micena fossils were dated to 1.3 million years, an age that far exceeded the limit of detectability of DNA in bones, so the likelihood of finding this biomolecule was slim. Attempts to amplify autochthonous DNA from equid fossils found at Venta Micena equivalent in age to the hominin materials from this site were unsuccessful. An alternative approach based on the prolonged survival of proteins in fossil bones (as noted above) was to analyze these biomolecules. Venta Micena fossil proteins were investigated by two independent groups: Lowenstein et al. at the University of California, San Francisco, and Olivares et al. at the University of Granada, Spain. The San Francisco team used RIA for the immunological detection of fossil proteins, while the Granada group used enzyme-linked immunosorbent assay (ELISA), a technique equivalent in sensitivity and detectability to RIA [42].

#### 4.2.1. Preparation of Fossil Extracts

Aseptic conditions were rigorously maintained, and disposable materials were used to avoid external contamination during extraction and testing processes. The fossil sample was ground to a fine powder and treated successively with phosphate-buffered saline (PBS), EDTA, and acetic acid. The supernatant was collected after each treatment (Figure 1).

#### 4.2.2. Immunoassays for the Detection of Albumin

Fossil extracts were placed in the wells of a plastic microtiter plate. During this process, some proteins became irreversibly attached to the plastic. In the ELISA, appropriate mouse antiserum or monoclonal antibody to specific albumin was added, and the second antibody used was a biotinylated goat antimouse immunoglobulin, followed by extravidin-peroxidase. Extravidin binds biotin, and peroxidase catalyzes a color reaction that can be quantified in a microplate autoreader. In the RIA, rabbit antisera were added to various species of albumins, followed by the second antibody, ^125^I-labeled goat antirabbit immunoglobulin. Radioactivity was quantified in a scintillation counter. Absorbance (ELISA) or radioactivity (RIA) obtained with fossil extracts was compared to the calibration curves for native albumins from different species in order to quantify the albumin detected.

#### 4.2.3. Detection of IgG by Quantitative Dot-Blotting

Fossil extracts were placed on a nitrocellulose membrane, which has a higher capacity than plastic to attach proteins. The appropriate anti-IgG antibodies labeled with peroxidase were then added. We then cut out the nitrocellulose circles within which the reaction took place, and transferred each circle to a microtiter plate well. Peroxidase catalyzes a color reaction that can be quantified in a microplate autoreader.

### 4.3. Detection of Albumin in Fossil Bones

Both the Granada and the San Francisco groups studied Venta Micena fossils VM-0 and VM-1960 attributed to hominins, along with fossil CV-1, a fragment of humerus found at the Cueva Victoria site in the province of Murcia, Spain, and dated to an estimated 0.9 million years [42,43]. For comparison, fossils of different mammals from Venta Micena and Cueva Victoria were also analyzed by the two groups. An extract from fossil VM-0 was tested with antisera against albumin from different species, and both groups found greater reactivity with antisera against human albumin, whereas reactivity with other antibodies, especially antihorse albumin, was much lower (Figure 2). The conclusion, therefore, was that the albumin detected in VM-0 was closer to that of humans than other species. Lowenstein and colleagues also detected collagen and transferrin with immunological reactions similar to human proteins in fossil VM-0. Samples from the VM-1960 humerus, also attributed to a hominin, yielded results similar to those for VM-0 at both laboratories. However, neither group detected albumin in CV-1, a hominin fossil from the Cueva Victoria site. In their studies of other mammals, both groups observed reactions similar to horse albumin in equid fossils, and the San Francisco group detected reactions similar to bison albumin in two bovine fossils. Together, these results confirm the presence of albumin and other proteins in fossils believed to be approximately 1.3 million years old, and demonstrated that it is possible to identify species-specific characteristics in these fossil proteins with immunological techniques [42].

### 4.4. Fossil Proteins or Contamination

One of the most important considerations in molecular paleontology is to determine whether the biomolecules detected are an integral part of the fossil (endogenous) or originate from exogenous contamination. Special protocols for fossil sample preparation are required to verify the endogenous or exogenous origin of the biomolecules (Figure 1).

Although the possibility of contamination was unlikely (for example) in the case of horse albumin detected in the equid fossils, the albumin found in VM-0 and VM-1960 could have originated from contact with sweat or saliva during handling by paleontologists. It has also been suggested that seepage from recent human remains may have contaminated the fossils [44]. This latter possibility was easily ruled out, since soil collected from the site where the hominin fossils were found was analyzed and no albumin was detected (Figure 3).

The fossil proteins were bound to the mineral phase of the bone, from which they were released by treating the samples with a decalcifying EDTA solution [42]. Proteins from exogenous contamination are not bound to the mineral phase and can therefore be extracted without dissolving the bone, i.e., by simply washing the sample in phosphate-buffered saline solution (PBS). Albumin not bound to the mineral phase can be detected in fresh, surgically removed human bone that contains albumin from retained blood, which is also easily eluted with PBS. Albumin not bound to the mineral phase has been detected in human bones that were buried for as long as 10 years (Figure 3) [45]. Fossils VM-0 and VM-1960 did not contain unbound albumin, and this protein was detected only when the mineral phase was dissolved with EDTA. Therefore, the albumin in VM-0 and VM-1960 was integrated into these fossils and did not arise from exogenous contamination. Given the endogenous origin of this albumin and its immunological reactivity similar to modern human albumin, these results support the ascription of both fossils to a hominin [38,42,45].

Subsequent criticism of these data was based on the work of Cattaneo et al. [46]. These authors, in an effort to reevaluate the results obtained by different groups that reported the detection of proteins in fossil bones and archaeological artefacts, buried recent human and bovine bones in garden soil, and reported that under these conditions, no albumin was detectable after 1 month in human bone and after 3 months in bovine bone. Palmqvist, based on these results, inferred that it is not possible to detect proteins in bones beyond a few months [47]. In connection with efforts to detect ancient DNA, Poinar pointed out that “There are many types of fossilization processes, and to assume that the breakdown of DNA is similar in all of them, or is equivalent to that in non-fossilized material, is not scientific” [48]; this reasoning may also apply to fossil proteins. A bone becomes a fossil only after being subjected to particular conditions over a prolonged period—conditions that are not replicated by burying fresh bones in garden soil for several months. The Venta Micena site, located near the shore of an ephemeral alkaline lake, displays very specific preservation conditions in which mammal bones were covered and buried in an impermeable deposit of carbonate mud shortly after the animals’ death. This mud consisted of calcite crystals that formed a film around the bone surfaces and protected them from alteration until complete burial and later excavation [49].

To shed further light on the fate of proteins in fresh buried and fossilized bone, Lowenstein and colleagues compared the amount of albumin detected in fresh human bone, human bone buried in a cemetery for 10 years, and in fossils VM-0 and VM-1960. Their findings showed that while the amounts of albumin detected in fossils were logically much lower than those found in fresh bone, the amounts of albumin detected in bone from the 10-year-old cemetery burial were not much higher than those observed in the fossils [45] (Figure 3). This apparent contradiction can be explained considering that after death, under normal conditions all proteins in most bones tend to disappear within a relatively brief period. However, under special conditions that lead to fossilization, such as those at the Venta Micena site, the proteins are “frozen” in the mineral phase of the bone and can be preserved for millions of years [24,38].

### 4.5. Monoclonal Antibodies to Study the Integrity of Fossil Proteins

Despite their persistence, fossil proteins are inevitably fractionated or denatured, although some amino acid sequences detectable with antibodies may survive. In the studies discussed above, antibodies obtained from the blood serum of an animal that was previously immunized against a protein (antiserum) were used. This antiserum contained a mixture of different polyclonal antibodies, each of which recognized an independent part (epitope) of the protein used for immunization. For more fine-grained molecular studies, monoclonal antibody technology can be used to produce a single type of antibody that reacts with a single epitope of the protein [50]. By independently testing different monoclonal antibodies against human albumin, it is possible to analyze different epitopes of this molecule individually, and to determine which of them have survived in the fossil albumin. In studies of the reactivity of monoclonal antibodies against human albumin with extracts of the hominin fossils VM-0 and VM-1960, it was found that each of the monoclonal antibodies showed a different degree of reactivity (Figure 4). Higher or lower reactivity indicated a better or worse degree of preservation of different epitopes recognized in the fossil albumin. Monoclonal antibodies, therefore, not only made it possible to confirm the data obtained with polyclonal antibodies, but also provided an opportunity to analyze the integrity of different epitopes of the protein [42,45].

If a polyclonal antiserum is used, the fractionated, degraded and denatured fossil protein, which has lost part of its epitopes, will capture fewer antibodies than the native protein used to generate an ELISA or RIA calibration curve. Therefore, for an equivalent number of molecules, the signal produced by the antiserum in reaction with the fossil protein will be weaker than with the native protein. However, if the epitope recognized by a monoclonal antibody is preserved in the fossil protein, the antibody will produce a signal with the fractionated fossil protein (lower molecular weight) that is quantitatively equivalent to that produced with the native protein (higher molecular weight) used for the ELISA or RIA calibration curve (Figure 5). Consequently, when the signals are situated on the calibration curve in order to quantify the amount of fossil protein, higher amounts than expected may be detected (Figure 5). For this reason, the term ng-equivalent was coined to indicate that the amounts of fossil protein detected in assays with monoclonal antibodies reflect the reactivity of each monoclonal antibody rather than the actual amount of fossil protein present in the sample [42,45,51]. In fact, when different monoclonal antibodies recognizing different epitopes of human albumin were used, the quantification of fossil albumin varied depending on the reactivity of each monoclonal antibody, thus reflecting the presence or absence of each recognized epitope (Figure 4). Thus, the use of monoclonal antibodies can evidence the greater or lesser preservation of different epitopes recognized in fossil proteins [45].

### 4.6. Proteins Other than Collagen in Fossils

A novel aspect of research with the Venta Micena specimens was the detection of albumin, a protein detected for the first time in such ancient fossils. Although collagen and osteocalcin, the most abundant proteins in bone, had previously been identified in fossil bones that were millions of years old [22,23,24], in principle the detection of albumin seemed unlikely since this protein is much less abundant in bone than collagen or osteocalcin. Moreover, because this protein is highly soluble, it was assumed that it would be rapidly washed out of bones during the degradation process. However, as Tuross and colleagues pointed out, the key phenomenon of protein preservation, i.e., the encapsulation of these biomolecules in hydroxyapatite crystals, appears to especially affect plasma proteins [52,53]; this could explain why albumin and other plasma proteins remained detectable in the Venta Micena fossils. Albumin, in some cases, becomes so highly concentrated in bone after death that it can reach levels higher than those found in living animals [52]. However, collagen—although abundant in bone—is an uninformative molecule in genetic terms because its amino acid sequence is highly repetitive and similar across different species. In contrast, albumin provides more evolutionary information. The fact that this protein has evolved more rapidly than collagen makes albumin a better discriminator between species [54]—clearly an advantage in paleoproteomics. It is thus not surprising that this protein has been studied with immunological methods in several extinct species such as the mammoth, the Steller elephant seal, the Tasmanian sea lion, and the quagga, in which the results have helped to resolve controversies regarding their phylogenetic affinities [55,56,57,58]. The DNA of some of these species was later sequenced, yielding phylogenetic trees very similar to those derived from the immunological study of albumin [59].

Another plasma protein detected in Venta Micena fossils was immunoglobulin G (IgG). Extracts from fossil equid bones and from fossil bones ascribed to hominins were tested with anti-human IgG and anti-horse IgG polyclonal antisera (Figure 6). The equid fossils showed higher reactions with anti-horse IgG than with anti-human IgG, while the hominin fossils reacted more strongly with anti-human IgG than with anti-horse IgG. These results demonstrate the feasibility of detecting species-specific IgG in different types of fossils. Samples from VM-1960, which had shown a stronger reaction to anti-human albumin, also showed a more marked reaction to anti-human IgG, whereas Cueva Victoria fossils, in which no albumin was detected [42] (Figure 3), showed no reaction to anti-IgG sera [51]. In parallel with the results of assays with albumin, no IgG was found when fossil samples were washed with PBS. The presence of IgG was seen only after decalcification with EDTA—a result interpreted to show that the proteins detected were embedded in the mineral phase (Figure 6) [42,51]. Although some paleontologists have denied the presence of hominins at the Venta Micena site based on morphological and other criteria (but have thus far not used molecular methods themselves) [35], the immunological findings in Venta Micena fossils strongly support the assignment to hominins of cranial fragment VM-0 and the two humeral fragments VM-1960 and CV-3691. Phillip V. Tobias, who described *Homo habilis*, opined: “The convergence of the two laboratories, working independently of one another, by two different methods, provides very strong evidence in support of the conclusion that all three of the bones from Venta Micena are of hominid origin. When the molecular and the skeletal data are considered together, the picture afforded by the bio-anthropological evidence is that the three bones are of human origin” [41].

The presence of proteins other than collagen in fossils is consistent with microscopic observations of cells and tissues in dinosaur bones [60,61]. Mongelard detected albumin with immunological methods in 1700-year-old rodent fossils [62], and this protein was recently identified with MS methods in different hominin fossils [63] and in a mammoth femur [64]. Furthermore, extracts of these fossils can be used to immunize animals as a way to obtain anti-fossil protein sera that can be tested with native proteins to determine protein and species specificities. For example, rabbit anti-Ramapithecus bone extract reacted with sera from different modern primates [56], rat anti-dinosaur bone extract reacted with avian and mammalian hemoglobin [65], and rabbit anti-sauropod eggshell extract reacted with chicken ovalbumin [66]. Similarly, antisera were obtained in mice against VM-0 and against a fragment of Neanderthal humeral diaphysis (fossil CU-1) found at the Cueva Umbría in Orce, Granada (50,000–70,000 years old). Anti-VM-0 serum showed higher reactivity with human albumin than did non-immune serum, and weaker albeit positive reactivity with human IgG. In related work, anti-Neanderthal serum reacted more strongly than anti-VM-0 serum with human albumin, IgG, hemoglobin and transferrin [67].

Although twentieth-century paleoproteomics research used less powerful technologies and less stringent protocols compared to currently available tools to assess contamination, confirm endogeneity, and authenticate species assignment, the body of knowledge provided then has paved the way for the extraordinary development of twenty-first-century paleoproteomics. However, peptides extracted from fossils may be altered, and this limits the applicability of MS methods [68]. In these cases, immunological methods hold potential to aid in efforts to increase the accuracy and reliability of fossil protein identification.

### 4.7. Twenty-First-Century Paleoproteomics and the Venta Micena Fossils

Paleoproteomics is a relatively young yet rapidly growing field of molecular science in which proteomics-based sequencing technology is used to identify species and propose evolutionary relationships among extinct taxa. As a complementary approach to paleogenomics, the study of ancient proteins has the potential to disclose older, more complete phylogenies due to the relative stability of amino acids in proteins compared to nucleic acids in DNA [69]. Mass spectrometry provides unprecedented information on modern and ancient proteomes, and can yield protein sequence data from extinct organisms as well as historical and prehistorical artifacts [70]. Since the seminal application of MS in paleoproteomics [71], extraction and analysis protocols, software for data processing, protein databases, and high-mass-accuracy instrumentation have all seen significant progress. These advances are potentially applicable to the study of Venta Micena fossils. Given the ages of these fossils—older than 1 million years—this approach to research can be considered deep time paleoproteomics [72]. Another promising approach to extend research on plasma proteins such as albumin or IgG is affinity purification coupled with MS, which has been used to selectively identify proteins [73,74]. In addition, antibodies to specific protein or peptide components can be used to detect and separate proteins from a heterogeneous mixture, making them ideal tools to study degraded or fragmented ancient proteins (targeted paleoproteomics) [72]. Furthermore, since collagen has been detected by immunological methods in VM-0, this fossil, along with the other Venta Micena fossils assigned to hominins, could also be analyzed with zooarchaeology by mass spectrometry (ZooMS), an efficient proteomics-based method of species identification by collagen peptide mass fingerprinting. This method is especially useful to determine the affiliation of fossils that are morphologically controversial, such as VM-0 [75]. Combining previous knowledge in the immunodetection of fossil proteins with recent advances in MS approaches will make it possible to access the great store of potential information locked in the Venta Micena fossil record, and will contribute to the growth of paleoproteomics.

## 5. Conclusions

In paleoproteomics research, immunological techniques predominated during the twentieth century, whereas the twenty-first century has seen significant developments and improvements in the application of more informative MS methods. Immunological methods have been used to obtain molecular data that shed light on the species ascriptions of the Venta Micena fossils. These methods showed that the fossils contained well-preserved serum proteins, i.e., albumin, IgG, and transferrin. This research also demonstrated the species specificity of fossil proteins, thus helping to resolve scientific controversies that arose regarding the morphological data used to ascribe the Orce skull and other Venta Micena fossils to hominins. The use of monoclonal antibodies that target human albumin made it possible to determine the degree of preservation of different epitopes recognized in the fossil albumin. The combination of immunological methods with antibodies together with MS (targeted paleoproteomics) could represent an important advance in the study of these fossils by providing data that can help determine the phylogenetic relationships between these fossils and other known species of hominins.

## Figures and Tables

**Figure 1 biology-11-01184-f001:**
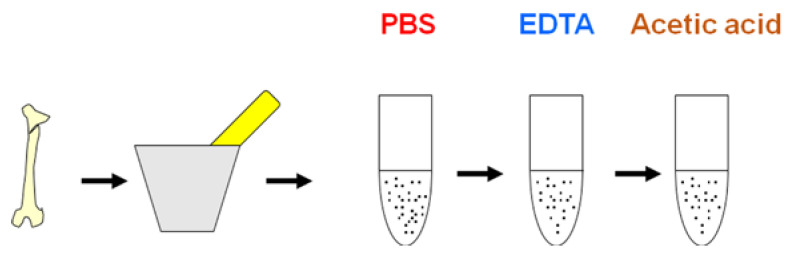
Preparation of fossil extracts.

**Figure 2 biology-11-01184-f002:**
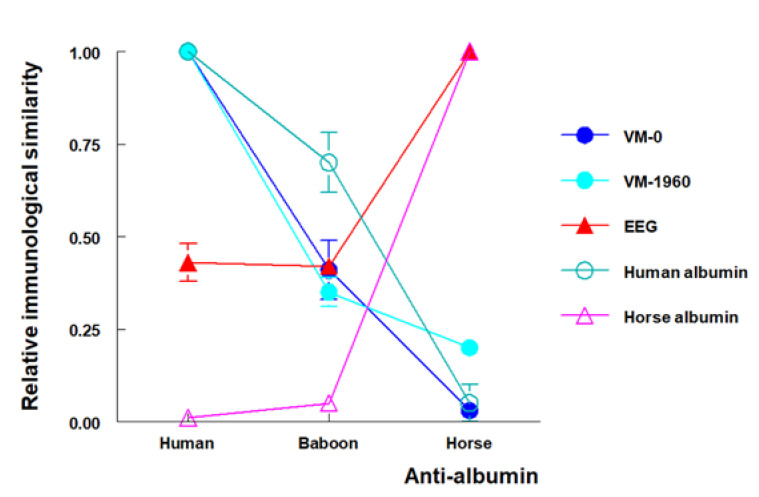
Reactivity of EDTA extracts of fossils from the Venta Micena site with three polyspecific mouse sera against human, baboon, and horse albumin in an enzyme-linked immunosorbent assay. VM-0 (skull fragment) and VM-1960 (humeral diaphysis) were assigned to hominins; EEG is a skull fragment of an equid fossil. All three specimens were dated to 1.3 million years old. The results are expressed as relative immunological similarity, defined as the ratio of each reaction to the homologous (most specific) albumin determination. VM-0 and VM-1960 produced a pattern of reactions similar to modern human albumin, whereas EEG produced a pattern of reactions similar to modern horse albumin. From Borja et al. [42], with permission.

**Figure 3 biology-11-01184-f003:**
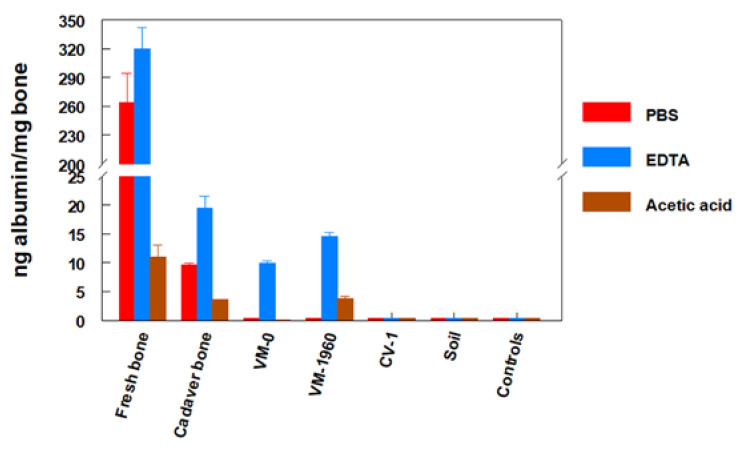
Enzyme-linked immunosorbent assay quantification of albumin in the PBS, EDTA and acetic acid extracts of human fresh bone (a fragment of femoral diaphysis), cadaver bone (a fragment of human occipital buried for approximately 10 years), fossils assigned to hominins (VM-0, VM-1960, CV-1), and soil collected around VM-0. Also shown are the buffers used for extractions, tested in the absence of bone. A mouse anti-human albumin polyclonal antiserum was used in all assays. From Lowenstein et al. [45], with permission.

**Figure 4 biology-11-01184-f004:**
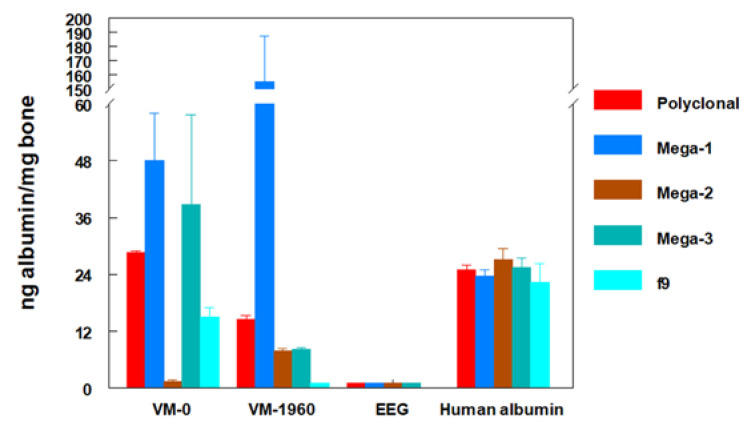
Enzyme-linked immunosorbent quantification of albumin in EDTA extracts of fossils VM-0 and VM-1960, assigned to hominins, and fossil EEG, assigned to an equid, with monoclonal antibodies. Four monoclonal anti-human albumin antibodies (MEGA-1, MEGA-2, MEGA-3 and 8F6F9) were used with the fossils. Polyclonal mouse anti-human albumin serum was used for comparison. Human albumin (25 ng) was tested for comparison. From Lowenstein et al. [45], with permission.

**Figure 5 biology-11-01184-f005:**
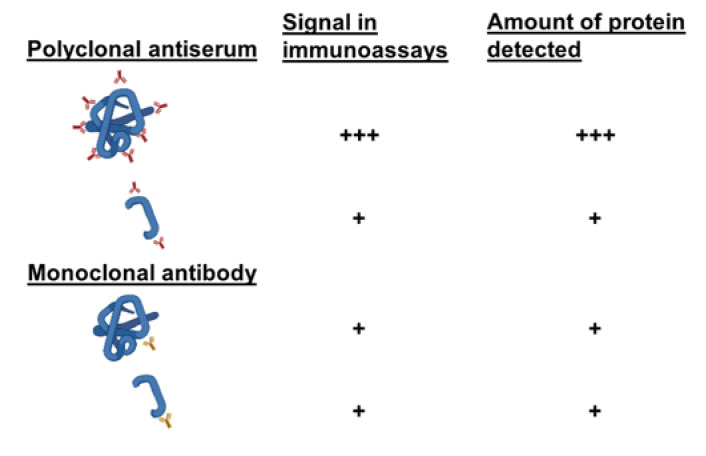
Differences in reactivity and quantification of fossil proteins (short peptide) in comparison to the native protein in assays with a polyclonal antiserum or a monoclonal antibody. Figure created in BioRender.com.

**Figure 6 biology-11-01184-f006:**
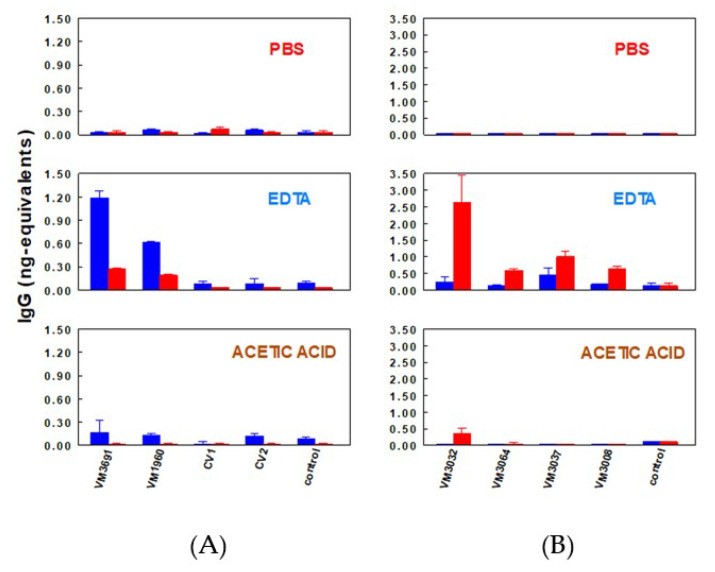
Quantities of human IgG (blue bars) and horse IgG (red bars) detected in PBS, EDTA and acetic acid extracts of (**A**) hominin and (**B**) equid fossils from Venta Micena, determined with quantitative dot-blotting. The extraction solutions were used as negative controls. From Torres et al. [51], with permission.

## Data Availability

Not applicable.

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
