# Peer review of "Twentieth-Century Paleoproteomics: Lessons from Venta Micena Fossils"

_biology, 2022, doi:10.3390/biology11081184_

Round 1

Reviewer 1 Report

In this manuscript, Torres et al. give a nice overview of twentieth century ancient protein studies with antibodies and specifically centered it on a case study using immunological techniques for species identification of hominid remains from Spain. I think this is a well written manuscript and I appreciate the short final discussion of twenty-first century paleoproteomics with mass spectrometry.

In the Twenty-first century section “Schroeter et al. 2022” should be a number.

Also, please use colorblind accessible palettes for the figures. Red and green together are difficult to distinguish with certain types of colorblindness. You can check your figures with the following website: https://www.color-blindness.com/coblis-color-blindness-simulator/

Author Response

In the Twenty-first century section “Schroeter et al. 2022” should be a number.

REPLY. Corrected.

Also, please use colorblind accessible palettes for the figures...

REPLY. We have changed the figures so that that red and green elements are not adjoining within the same figure.

English language and style...

REPLY. Corrected.

Reviewer 2 Report

The manuscript of Torres et al. is a well-structured overview of paleoproteomics methods applied to the 1.3 million years old human fossils discovered in the Orce settlement (Granada, Spain) to resolve the controversies related to their origin. The review addresses all the critics that the studies of those fossils faced from the moment they were described as human remains (1989) and provides a valuable discussion of challenges and caveats that paleoproteomics has been facing over time.

There are some minor suggestions that I would like to make:

Lines 194-203: it is not clear to which publications of the two independent groups the authors refer. Are they several or only one (42- Borja et al. 1997)? It would be convenient to make a more clear statement from the beginning of which publications are reviewed. 

Line 255: exact citation is needed along with Cattaneo et al., which is 46.

Lines 281-282: Such a strong statement needs a reference.

Line 405: put the abbreviation along with the full name "Mass spectrometry (MS)", as you use only the abbreviation later (for example, in line 407).

Line 417: reference (Schroeter et al., 2022) is not numbered in the bibliography.

Author Response

Lines 194-203: it is not clear to which publications...

 REPLY. Reference 42, whose object was the identification of VM-0 and VM-1960, is a joint publication of the two independent groups. This reference has been placed at the beginning of the comments on Venta Micena fossils in the second version of the manuscript, and is associated with Figure 2, as indicated in the figure legend. The 2 groups also jointly published reference 45 associated with Figures 3 and 4 (see also the legend to these figures). This reference is included below, when commenting on aspects such as contamination and the use of monoclonal antibodies. Reference 51 on IgG was published only by the Granada group.

Line 255: exact citation is needed along with Cattaneo et al., which is 46.

REPLY. Corrected

Lines 281-282: Such a strong statement needs a reference.

REPLY. Two references have been added.

Line 405: put the abbreviation...

REPLY. Done.

Line 417: reference (Schroeter et al., 2022) is not numbered in the bibliography.

REPLY. Corrected.

Reviewer 3 Report

This is an interesting review paper that used the example of the Venta Micena hominin to discuss the use of paleoprotomics in paleontology. 

This paper warrants publication, but it needs to be organized before publication. There should be more detail about the methodologies so readers with less experience in proteomics can appreciate the paper. On the other end, the conclusions need to be stronger. What is the impact of these studies on our understanding of VM-O as a hominin? Has it settled the discussion? It is unclear from the current text. 

All the figures need to be enhanced with better resolution. Currently, they are difficult to read. 

There needs to be further discussion on ZooMS, developed extensively in the 21st century and applied to Holocene zoological assemblages and many Paleolithic hominins such as the Denisovans (Brown et al., 2016).

The paper needs moderate English editing. 

Author Response

This paper warrants publication, but it needs to be organized...

REPLY. A new section with methodologies has been added (4.2. Methods to investigate proteins in fossils). The conclusion based on immunological data for the hominin assignation of fossils from Venta Micena is on page 10, lines 403-411. We have added Phillip V. Tobias’ opinion on our studies to strengthen this conclusion. At present the discussion about Venta Micena fossils is ongoing. Although the immunological results, which were obtained by two independent groups that used two different methods, are clear and objective, and no molecular research to date has discredited these results, some paleontologists remain convinced that there are no hominins at the Venta Micena site (see ref. 35).

All the figures need to be enhanced with better resolution.

REPLY. We have increased the resolution of the figures.

There needs to be further discussion on ZooMS...

REPLY. ZooMS has been mentioned in the revised Discussion (page 11, lines 459-465).

The paper needs moderate English editing.

REPLY. The manuscript was edited by a native-English-speaking editor and translator with decades of experience.